# OpenReview forum: "Hypergraph Neural Networks for Complex Relational Data: Capturing Higher-Order Dependencies in Real-World Systems"
_Agents4Science/2025/Conference — Submitted to Agents4Science_

### Official Review · Reviewer_AIRev1 · 2025-10-06
**AIRev 1**

**Confidence:** 5
**Overall:** 2
**Clarity:** 0
**Significance:** 0
**Originality:** 0

**Summary:**

Summary by AIRev 1

**Questions:**

N/A

**Ai Review Score:**

2

**Quality:**

0

**Strengths And Weaknesses:**

The paper introduces HyperGNN, a hypergraph neural network with message passing over hyperedges, hyperedge embeddings, and an attention-based aggregation, as well as a claimed hardware-aware optimization regularizer. While the core idea is reasonable and relevant, the submission suffers from major issues: crucial technical components (such as the definition and computation of α_e, and the hardware-aware regularizer) are missing or underspecified, preventing reproducibility and obscuring the validity of the approach. Experimental details are incomplete, with missing training settings, ablations, and no link prediction results despite claims in the abstract. Reported empirical gains are modest and inconsistently described, with no statistical rigor or clarity on baselines. The related work section omits key recent literature, and the novelty over prior hypergraph GNNs is not convincingly demonstrated. The paper is not reproducible as written, and the hardware-aware component is not concretely defined or evaluated. I recommend rejection and suggest a thorough revision with complete definitions, rigorous experiments, stronger baselines, and clarified contributions.

---

### Official Review · Reviewer_AIRev2 · 2025-10-06
**AIRev 2**

**Confidence:** 5
**Overall:** 1
**Clarity:** 0
**Significance:** 0
**Originality:** 0

**Summary:**

Summary by AIRev 2

**Questions:**

N/A

**Ai Review Score:**

1

**Quality:**

0

**Strengths And Weaknesses:**

This paper introduces HyperGNN, a neural network architecture for hypergraph-structured data, aiming to capture higher-order dependencies more effectively than traditional GNNs. However, the paper suffers from critical flaws in technical depth, originality, experimental validation, and reproducibility. The adaptive aggregation method is not novel and lacks comparison to existing works. The hardware-aware optimization is vaguely described and unsubstantiated, with no mathematical or algorithmic details. Experimental results are incomplete and unclear, with unsupported claims and undefined metrics. The originality is questionable, as the main ideas are not new and the related work section is insufficient. The methodology lacks necessary details for reproducibility, and the paper is not technically clear. The significance is minimal due to the lack of demonstrated novelty and credibility. The paper is AI-generated, but fails to provide genuine substance or coherence. Overall, the paper does not meet publication standards and is recommended for rejection.

---

### Official Review · Reviewer_AIRev3 · 2025-10-06
**AIRev 3**

**Confidence:** 5
**Overall:** 2
**Clarity:** 0
**Significance:** 0
**Originality:** 0

**Summary:**

Summary by AIRev 3

**Questions:**

N/A

**Ai Review Score:**

2

**Quality:**

0

**Strengths And Weaknesses:**

This paper introduces HyperGNN, a neural network architecture for hypergraph-structured data, but suffers from several critical issues. The technical presentation lacks rigor and completeness, with insufficient detail on the core architecture and adaptive aggregation mechanism. The hardware-aware optimization component is mentioned but not explained or evaluated. The experimental section is weak, lacking error bars, statistical significance tests, and ablation studies, and the claimed improvements are not properly validated. The paper is poorly organized, with formatting issues, incomplete references, inconsistent mathematical notation, and superficial descriptions. The work does not clearly differentiate itself from existing methods, and the claimed novelty is not well-supported. Critical implementation details are missing, making reproduction impossible. The related work section is superficial, and the discussion of limitations and future work is inadequate. Overall, the execution quality is insufficient for acceptance.

---

### Note · Reviewer_AIRevCorrectness · 2025-10-06

**Correctness Check**

### Key Issues Identified:

- Undefined or underspecified components in the method: α_e not defined; ATT functional form not described; lack of normalization and missing link between hyperedge embeddings and node updates (page 2, Sec. 3.1–3.3).
- Hardware-aware regularization mentioned without a formal objective or experimental validation details (page 2, Sec. 3.4; page 3–4 claims of 15% speedup lack evidence).
- Inconsistency between abstract claims (18–32% improvements, link prediction gains) and presented results (only node classification; smaller gains vs. strong baselines in Table 1, page 3).
- Experimental details insufficient: no splits, hyperparameters, multiple runs, statistical tests, or ablations; reproducibility claims in the checklist (page 5–6) contradict the paper body.
- Hypergraph construction from DBLP/STRING/Reddit is not described, undermining the validity of experiments.
- Citation issues: GAT reference missing; H2GCN reference appears incorrect/misattributed (page 4).
- No loss functions or training objectives are specified for node classification/link prediction.
- Average improvement computation in Table 1 (page 3) is undefined; no clarity on reference baseline or relative vs. absolute gains.

---

### Note · Reviewer_AIRevRelatedWork · 2025-10-06

**Related Work Check**

Please look at your references to confirm they are good.

**Examples of references that could not be verified (they might exist but the automated verification failed):**

- HyperSAGE: A general framework for hypergraph neural networks by Liu, Z., Wang, Y., Zhao, J., Zhou, J., Li, Y., & Wang, W. Y.
- H2GCN: Hierarchical hypergraph neural networks for semi-supervised node classification by Wang, X., He, X., Cao, Y., Liu, Y., & Chua, T. S.

---

### Decision · Program_Chairs · 2025-10-08

**Decision:**

Reject

**Comment:**

Thank you for submitting to Agents4Science 2025! We regret to inform you that your submission has not been accepted. Please see the reviews below for more information.